# Nanotubes Formation in *P. aeruginosa*

**DOI:** 10.3390/cells11213374

**Published:** 2022-10-26

**Authors:** Faraz Ahmed, Zulfiqar Ali Mirani, Ayaz Ahmed, Shaista Urooj, Fouzia Zeeshan Khan, Anila Siddiqi, Muhammad Naseem Khan, Muhammad Janees Imdad, Asad Ullah, Abdul Basit Khan, Yong Zhao

**Affiliations:** 1College of Food Science and Technology, Shanghai Ocean University, Shanghai 201306, China; 2Microbiology Analytical Centre, FMRRC, PCSIR Laboratories Complex, Karachi 75280, Pakistan; 3Dr. Panjwani Center for Molecular Medicine and Drug Research (PCMD), International Center for Chemical and Biological Sciences, University of Karachi, Karachi 25270, Pakistan; 4Department of life Science, Shaheed Zulfiqar Ali Bhutto Institute of Science & Technology, Karachi 75600, Pakistan; 5Department of Microbiology, University of Karachi, Karachi 25270, Pakistan; 6Department of Pathology, Dow University of Health Sciences, Karachi 74200, Pakistan; 7Laboratory of Quality & Safety Risk Assessment for Aquatic Products on Storage and Preservation (Shanghai), Ministry of Agriculture, Shanghai 201306, China; 8Shanghai Engineering Research Center of Aquatic-Product Processing & Preservation, Shanghai 201306, China

**Keywords:** adhesion, biofilms, nanotubes, pseudomonas, small colony variants

## Abstract

The present study discusses a biofilm-positive *P. aeruginosa* isolate that survives at pH levels ranging from 4.0 to 9.0. The biofilm consortia were colonized with different phenotypes i.e., planktonic, slow-growing and metabolically inactive small colony variants (SCVs). The lower base of the consortia was occupied by SCVs. These cells were strongly attached to solid surfaces and interconnected through a network of nanotubes. Nanotubes were observed at the stationary phase of biofilm indwellers and were more prominent after applying weight to the consortia. The scanning electron micrographs indicated that the nanotubes are polar appendages with intraspecies connectivity. The micrographs indicated variations in physical dimensions (length, width, and height) and a considerable reduction in volume due to weight pressure. A total of 35 cells were randomly selected. The mean volume of cells before the application of weight was 0.288 µm^3^, which was reduced to 0.144 µm^3^ after the application of weight. It was observed that a single cell may produce as many as six nanotubes, connected simultaneously to six neighbouring cells in different directions. The in-depth analysis confirmed that these structures were the intra-species connecting tools as no free nanotubes were found. Furthermore, after the application of weight, cells incapable of producing nanotubes were wiped out and the surface was covered by nanotube producers. This suggests that the nanotubes give a selective advantage to the cells to resist harsh environmental conditions and weight pressure. After the removal of weight and proper supply of nutrients, these phenotypes reverted to normal planktonic lifestyles. It is concluded that the nanotubes are not merely the phenomenon of dying cells; rather they are a connectivity tool which helps connected cells to tolerate and resist environmental stress.

## 1. Introduction

*Pseudomonas aeruginosa* (*P*. *aeruginosa*) is one of the most widely studied opportunistic pathogens among gram-negative bacteria [1]. It is capable of surviving in different environmental conditions including soil, water, hospital settings, food production units, etc. [1,2]. The survival of *P. aeruginosa* is attributed to its adaptability to different environmental conditions and resistance mechanisms [3]. According to the World Health Organization, *P. aeruginosa* is one of the major antibiotic-resistant pathogens [4]. Moreover, it is capable of surviving in acidic and alkaline conditions and tolerates high salinity, high sugar and desiccation for many days [5]. The long-term survival and high level of resistance of *P. aeruginosa* to antibiotics, disinfectants and environmental stress are due to biofilm lifestyle and metabolically inactive persister cells [2].

Biofilm formation is a major problem in hospital and industrial setups [4]. This is a protective consortium that shelters bacteria from invaders, immune responses and antibacterial agents. Additionally, biofilm is a multicellular aggregate comprising different phenotypes of single or multiple species [6]. In this consortium, each phenotype plays its role in long-term survival and persistence [2,6]. The small colony variants are responsible for cementing the consortium by producing higher amounts of the extracellular matrix material. The persister cells are hyper-adhesive and their planktonic form is responsible for colonization at different sites [7,8]. Increased resistance offered by biofilm indwellers to disinfectants and conventional cleaning methods is a serious threat to public health in industrial and clinical settings. These pathogens may persist in food processing plants and increase the risk of post-processing contamination [9]. In vivo and invitro studies have shown that *P. aeruginosa* isolates produce variations in colonial morphology in the solid medium [10]. Hyper-adhesive wrinkles and small colony phenotypes form a major part of the *P*. *aeruginosa* biofilm community [10]. It is reported that these phenotypes produce excessive amounts of exopolysaccharides which cause antibiotic tolerance, altered metabolism, reduced immunogenicity and increased persistence in biofilms [10,11]. Several studies have reported that *P. aeruginosa* adopts different mechanisms of biofilm formation [11]. A study [12] has suggested that extracellular macromolecules and appendages play an important role in bacterial responses to surface attachment. Flagella and pili are reported to be involved in the formation of mature multicellular structures in *P. aeruginosa* biofilms that may contribute to quorum sensing-controlled DNA release [13]. Oldewurtel and colleagues [14] have suggested that type 4 pili of *Neisseria gonorrhoeae* strongly interact with each other and restrict the swarming expansion rate and initiate multicellular aggregates and biofilm formation. Recently, another extra-cellular tube-like structure involved in bacterial communication in biofilms has been reported [15]. These structures are known as nanotubes, involved in bacterial communication and exchange of the DNA, protein and nutrients [15,16]. Recent research has suggested that nanotube-like structures were produced by multiple bacterial species attached to a solid surface or in biofilm communities [17]. Cao and colleagues [18] have shown that nanotube networks may aid cell–cell communication, thereby promoting biofilm development. The present study discusses the development of nanotubes in *P. aeruginosa*. The subject isolates were recovered from a sample with an acidic pH (4.5) and showed biofilm formation on glass slides. Nanotubes were first identified in *Bacillus Subtilis* and characterized as a mediator of cellular communication. The rationale for this claim was given by showing cytoplasmic fluorescent molecules transferring from one cell to another through these structures. Furthermore, these structures were found capable of transferring genetic material between two bacterial cells [19]. Another study showed that the formation of nanotubes is associated with the growing condition and growth phase of the organism, external stress applied and sample preparation methods, and genetic background. The study further discusses that the formation of nanotubes was exclusively attributed to the dying cells most likely due to biophysical stress. The rate of formation was monitored and found to be extremely fast. The study concludes that the formation of nanotubes is an extremely rapid response in dying cells under physical stress to transfer their cytoplasmic content. The present study presents the formation of extensive nanotubes in *P. aeruginosa* under the stress of applied external weight. Cells were shown to extrude more than one and up to six tube-like structures connecting with the neighbouring cells. Cells underweight were shown to be reduced in volume and the explanation could be an exchange of cytoplasmic content. This study warrants additional experimentation to confirm this phenomenon. It is the first report of extensive nanotube formation in *P. aeruginosa* under stress, and they are acting as a conduit for the transfer of cytoplasmic content [20].

## 2. Material and Methods

### 2.1. Bacterial Strains, Identification and Growth

Subject isolates of *P. aeruginosa* were recovered from a spice mix sample. Isolate’s growth was monitored on Pseudomonas Cetrimide Agar (Oxoid. Ltd. Hampshire, UK). Tryptone Soya Broth (TSB) (Oxoid. Ltd. Hampshire, UK) was used for primary enrichment and recovery of the subject isolate. Further confirmation was done by the amplification of *oprI* and *oprL* genes using specific primers, as described earlier [21].

### 2.2. Formation of Biofilm

The *P. aeruginosa* isolates were grown in TSB for 18 to 24 h at 35 °C. The cells were separated by centrifugation at 10,000 rpm and washed with phosphate-buffered saline (PBS) (pH = 7.4) and adjusted to a concentration corresponding to 0.5 McFarland standard (1.5 × 10^8^ CFU/mL). Tryptone soya broth (TSB) was prepared and distributed into 100 mL aliquots with different pH (4.0, 5.0, 7.0, 8.0 and 9.0). Subsequently, 100µL of the above cell suspension was inoculated in each flask (Final Count ≈ 1.5 × 10^8^ CFU/mL). A glass slide was placed in each flask to provide a base for the adhesion of cells and incubated at 25 °C, 35 °Cand 45 °C for 7 days [22]. Growth was monitored daily for 7 days. To study biofilms, the glass slides were carefully collected from the flask and washed with phosphate-buffered saline to clean the unbound debris. The biofilm was fixed in the microwave for 30 s at 450 Watts and stained with 0.1% crystal violet solution (20 min). The slides were washed again with PBS and biofilm-bound crystal violet was solubilized in 200 μL of an ethanol–acetone (4:1) mixture. The optical density (OD) of each slide was measured at 578 nm using a spectrophotometer (UV/Visible spectrophotometer, Shimadzu Corporation, Kyoto, Japan) [23]. Additionally, in a separate experiment, slides were horizontally placed in a flask and covered with another slide to study the effect of weight (15.45 g) on biofilm phenotype. *P. aeruginosa*, *S. aureus* and *E. coli* were used in this experiment to study inter- and intra-species nanotube formation. The experiment was repeated in triplicate.

### 2.3. Cell Segregation

After rigorous washing with PBS adherent biofilms on glass slides with weight pressure and at normal atmospheric pressure were detached in 5 mLPBS using an ultrasonic bath for 2 min at a cycle of 15 s. burst, 15 s. rest (Virtis Virsonic 300). The temperature was maintained at 22 °C in an ice bath [24]. The final concentration of each suspension was adjusted to 1 × 10^4^ CFU/mL corresponding to an OD value of 0.3 at a 579 nm wavelength. Samples at this point were collected for total RNA extraction for RT-PCR. Afterwards, 10-fold serial dilutions were made in sterile PBS tubes from each biofilm suspension and plated (100 μL) onto Pseudomonas Cetrimide Agar (Oxoid. Ltd. Hampshire, UK) and Tryptone Soya Agar (TSA). After incubation at 35 °C for 24 h, samples were collected and RT-PCR was performed for each sample adjusted to the same concentration (1 × 10^4^ CFU/mL) as before.

### 2.4. Cell Surface Hydrophobicity Assay

The hydrophobicity of strains was evaluated by the microbial adhesion to solvent test, as described in another study [25]. It consisted of evaluating the affinity of the cells towards a polar solvents (hexadecane). For the experiment, bacterial cells were harvested by centrifugation at 8500× *g* for 5 min and re-suspended in 0.01 M of potassium phosphate buffer to obtain an OD value of 0.13 against a wavelength of 578 nm. This bacterial suspension was mixed with a solvent in a ratio of 1:6 (0.4/2.4 *v*/*v*) by vortexing for 3 min to make an emulsion. The mixture was then left for 30 min until the separation of the two phases. Aqueous phase absorbance was measured (ABS 2) and the percentage of adhesion was expressed as % adhesion = (1 − ABS 2/ABS 1) × 100.

### 2.5. RNA Extraction and RT-PCR

RNA extraction was performed to observe the expression of *mreB* and *oprF* genes. These genes are responsible for the expression of outer membrane protein and maintaining the shape of an organism. RNeasy kit (QIAGEN, Venlo, The Netherlands) was employed to perform RNA extraction. Genomic DNA was eliminated by RQ1RNase-Free DNase (Promega). Real-Time PCR was performed on ABI 7500 Real-Time PCR machine (Applied Biosystem) with SYBR Green PCR Master Mix. The sequence of primers and PCR conditions for *mreB* and *oprF* genes were followed as described earlier [26].

*mreB_F*    5′-GGCTCGATGGTCGTAGACA-3′*mreB_R*    5′-ACGTAGGTGACGATGGCTTC-3′.*oprI_F*   5′-AGCAGCCACTCCAAAGAAAC-3′*oprI_R*   5′-CAGAGCTTCGTCAGCCTTG-3′.

Briefly, Initial denaturation was performed at 95 °C for 10 min followed by 40 cycles of 95 °C for 15 s and 60 °C for 1 min. PCR reactions were performed in quadruplicate for each gene and standard deviations were less than 0.15 Ct.

### 2.6. Effect of Triton X100 on biofilms

Biofilm-positive slides were washed with PBS and placed in 100 mL TSB flasks with and without Triton X100 and incubated at 35 °C for 24 h. The biofilm biomass was measured every 15 min for 3 h. The final biofilm biomass measurement was taken after 24 h. After incubation, slides were negatively stained with 2% (*w*/*v*) uranyl acetate. Electron microscopy was performed as described below. All observations are based on interpretations from 5 to 10 slides.

### 2.7. Electron Microscopy

Scanning electron microscopy was performed as described [20]. Briefly, biofilm-positive slides were divided into 4 mm sections and washed with distilled water to remove debris and negatively stained with 2% uranyl acetate for 30 s. Dehydration was carried out with ethanol (absolute) initially in 50% for 30 min, followed by 75% for 30 min and 95% for 30 min. After dehydration samples were platinum coated by the auto-fine coater (JEC-3000FC) at 20 mA current in a vacuum for 30 s.

Images were acquired using JSM IT 100 JEOL (Japan) Electron Microscope using the following parameters. The tungsten is an electron beam source, with high vacuum conditions for high-resolution and visualization purposes. The images were obtained at 5 to 20 KV electron volts (depending on the sample) at a working distance of 5–10 mm from the pole piece. Modes of imaging were acquired using Secondary electron detector (SEI) images.

### 2.8. Image Processing

All images were processed using Gwyddion 2.60. For each image, pixels were set at 1280 Px and 960 Px along the horizontal and vertical axis, respectively. All images were analyzed at unit channel G. The X and Y dimensions were adjusted according to the given scale for each image. Dimensions along the *Z*-axis for each sample unit were set at 1 µm, uniformly, throughout this analysis. Flattening of the surface slope and levelling of data by mean plain subtraction was performed by selecting the “levelling data by mean plane subtraction” command on the Gwyddion interphase. The command is guided under the equation Z (X, Y) = a +bx + cx+ dx+ By + Cy + Dy^3^.This is the least square root method for approximating surfaces with the plain. Next, levelling of data to arrange facet points upward was performed using the “facet point upward command”. Image plains of individual cells were made three points flat by applying the function “three-point levelling”. In some images, spikes were filtered using median filtration. Striated lines were removed by line-by-line levelling using the “align rows command” on the data process interphase. Line-by-line levelling may introduce some common artefacts which could be removed using the masking command provided in Gwyddion tools. The histogram adjust command was used to highlight the top features of the images. Quantitative physical characterization of surface features was performed using line profile command on tools interphase. This feature allows precise measurements of the particle’s height, width and length. Specialized functions present in the volume menu toolbox of GWYDDION 2.60 were explored to extract volume data as curves or spectra in each pixel. The software interprets data as a set of curves attached to each pixel in the XY pane when the stack of images along the Z axis is not available. Curves attached to each pixel in the XY plane act as an alternate special axis which corresponds to the Z axis.

### 2.9. Statistical Analysis

Data are represented by the column statistics of Mean and Standard deviation. Statistical differences between multiple physical parameters of two groups, cells under pressure and normal cells, are determined by using an unpaired two-tailed T-Test for distributions of volume, length, width and height (normally distributed), and the Wilcoxon signed rank test for the distribution of the number of nanotubes. The *p*-value < 0.05 was considered significant. Samples were randomly assigned, and the size of each column was taken as 35. All experiments were performed in triplicate.

## 3. Results

The *P. aeruginosa* isolates survived at pH ranges of 4.0 to 9.0 in TSB, though growth was slow at alkaline pH (9.0). The subjected isolate showed visible growth after 18 h of incubation at 37 °C, and the pH of the media was changed from alkaline to acidic (pH 6.5). The initial attachment of organisms to glass surfaces was weak, reversible and irregular. However, after 48 h of incubation, an irreversible adhesion was observed. The upper surface of the biofilm consortia was made up of loosely interconnected cells covered with the matrix material. The centre of the consortia was populated with strongly interconnected cells covered with matrix material and planktonic populations. The lower base of biofilm consortia was dominated by slow-growing and metabolically inactive cells or SCVs. This has already been reported in our previous studies [22,27]. An important observation of this study is the development of nanotubes in SCVs. These cells were strongly attached to the solid surface, as well as interconnected through a network of nanotubes (Figure 1, Figure 2, Figure 3 and Figure 4). This feature was more dominant in cells occupying the lower base of the biofilm consortia. After adhesion to the surface, another slide was placed on the consortia. The population between the two sides gave a brick-like appearance and cells rapidly became connected through a network of nanotubes (Figure 2). The atomic force microscopy image processing results have provided strong evidence for a variation in physical dimensions (length, width and height) and a considerable reduction in volume (Figure 5 and Figure 6) due to weight pressure. The mean volume of cells before weight application was 0.288 µm^3^, which was reduced to 0.144 µm^3^ after the application of weight. The difference was the statistically significant *p*-value < 0.0001 (Table 1 and Table 2) (Figure 5 and Figure 7). To compensate for the reduction in volume and increased internal pressure, cells tend to form nanotubes. Figure 5 and Figure 6 explain that height (*p*-value < 0.001) might be the major contributing factor in volume reduction. Under-weight stress, 77.14% of cells produced nanotubes while in the normal population only 28.57% of cells developed nanotubes (Figure 8a). The difference in the number of nanotubes per cell is also significant in the two groups (*p* value < 0.02); cells experiencing weight pressure tend to form four nanotubes, most of which (60.52%) are polar compared to normal cells, most of which do not develop nanotubes. It was observed that a single cell may produce as many as six nanotubes and could connect simultaneously to six neighbours in different directions (Figure 4c). The in-depth observations further revealed that these connecting structures “nanotubes” are intra-species connecting tools, as no signs of long extended and isolated nanotubes were observed. The size of the tubes appeared dependent on the distance of neighbouring cells (Figure 3 and Figure 4). The maximum length observed was 0.946 µm and the minimum was 0.336 µm (mean length 0.597 µm) (Table 1). The majority of the cells were found to harbour polar nanotubes (Table 1 and Table 2 and Figure 3). Around 72% of the nanotubes in a normal population and 60.5% in cells under pressure originated from poles (Figure 1, Figure 2 and Figure 8). Additionally, mid-cell connections were also found in *P. aeruginosa*, along with polar nanotubes (Figure 1 and Figure 2). It was observed when a single cell was connected to more than one cell (Figure 2a,b). The scanning electron micrographs indicated only the short intra-species nanotubes. There were no indications of long intercellular and extending nanotubes (Figure 1, Figure 2, Figure 3 and Figure 4). The slides without pressure were occupied by a mixed population i.e., cells with and without nanotubes (Figure 1a,b and Figure 8a). Moreover, after the application of pressure, most of the cells in biofilm consortia were dispersed and the cells connected with the nanotubes network persisted and survived (Figure 2). The removal of weight resulted in enhanced cell growth and biofilm biomass. This was confirmed by analysis of the sediment of cells formed at the bottom of the culture flasks, and clumps of cells in broth (Figure 5). In both situations, cells were devoid of nanotube structure. Similarly, colonies picked from the agar surface after 120 h of incubation did not show noticeable nanotube formation. However, after the application of the cover-slip (weight = 1.25 g), nanotubes were observed after 48 h of incubation in cells attached to the cover-slip. In this case, cells presented with normal *P. aeruginosa* morphology with slight variability in cell size. The size variation was also observed in normal biofilm consortia as well as in cells with nanotubes attached to the glass slides. The volume indicates a considerable variation in the size of the nanotubes.

### 3.1. The mreB and oprF Gene Expression in P. aeruginosa

The *mreB* gene is responsible for maintaining shape in many bacteria. The comparative analysis showed that *mreB* gene expression was decreased in cells attached to slides. The sub-culturing of these isolates on TSA resulted in significant increases in *mreB* gene expression. The results show that *mreB* expression gradually increases with incubation time. The *oprF* expression was low in cells attached to the slide. However, a significant increase in *oprF* expression was noticed after 1st sub-culture. The *OprF* and *mreB* expression was three-fold higher in cells after rejuvenation as compared to the biofilm population (Figure 6).

### 3.2. Cell Segregation

The cells with nanotubes were difficult to disperse from the surface even after shaking for 15 min at 200 rpm and sonication for 2 min at a cycle of 15-s burst, and 15-s rest. Therefore, these phenotypes were easily segregated from the rest of the biofilm consortia. They produced very small transparent pin-pointed colonies on TSA after 48 h of incubation. Colonial characteristics were maintained for the two subcultures. However, the cell shape reverted to normal *P. aeruginosa* after a single subculture.

### 3.3. Effect of Triton X100 on Biofilms

The nanotube-positive phenotypes were exposed to 10mg/L of Triton X100 for 30 s. It results in the complete disappearance of nanotubes and cells were squeezed within 30 s of exposure. However, the basic structure of the cell was intact in rod shape (Figure 9a,b).

### 3.4. Optical Density

Normal biofilm consortia showed gradual increases in biofilm OD as well as in extracellular matrix material production. Conversely, after the application of weight, a drastic reduction was noticed in biofilm OD (Figure 10).

### 3.5. Nanotube in Mix Culture of P. aeruginosa, E. coli and S. aureus 

To determine inter- and intra-species nanotube connections, a combination of *P. aeruginosa*, *E. coli* and *S. aureus* in biofilms was analyzed. All the subject strains showed nanotube formation after 48 h of incubation at 35 °C. Similar to *P. aeruginosa*, *E. coli* and *S. aureus* also showed nanotubes in cells at lower bases of biofilm consortia which were directly attached to glass slides. Cells of *S. aureus* were also compressed and appeared hexagonal, similar to *P. aeruginosa*. In mixed-species biofilm consortia, intra-species nanotube contact was common. Interestingly, no inter-species nanotube connection was observed among *P. aeruginosa*, *E. coli* and *S. aureus*. SEM images depicted that *P. aeruginosa* connected to *P. aeruginosa* via nanotubes, but not to *E. coli* or *S. aureus*. Similarly, *E. coli* and *S. aureus* also showed only intra-species nanotubes. The previous studies have shown the formation of inter-species nanotubes, contradicting previous reports the present study is devoid of these inter-species structures. One possible explanation could be the fact that the formation of nanotubes is dependent upon the phase and condition of bacterial growth, stress factors and the genetic background of the organism. These factors may have modulated the deviations in this process of inter-species nanotube formation.

### 3.6. Effect of Temperature Change on Nanotube Formation

The test isolate of *P. aeruginosa* showed biofilm formation at variable temperatures e.g., 25 °C, 35 °C and 45 °C. At 25 °C, the irreversible adhesion to glass slides was observed after 72 h of incubation. At 35 °C and 45 °C, strong and irreversible biofilm formation was achieved after 48 h of incubation. The nanotube formation was observed at 25 °C and 35 °C after the placement of another slide on biofilm consortia. At 45 °C, the nanotube structure was not found in *P. aeruginosa* isolates. The population analysis assay indicated that these phenotypes were comparatively slow growing and produced transparently and pinpointed colonies after 48 h of incubation at 35 °C. The numbers of these slow-growing colonies increase with biofilm optical density. Conversely, the number of wild-type *P. aeruginosa* colonies decreased after the appearance of slow-growing colonies in biofilm consortia. Moreover, cell surface hydrophobicity also increased with biofilm optical density and slow-growing colony variants.

### 3.7. Cell Surface Hydrophobicity before and after Nanotube Development

The hexane binding assay indicates that cell surface hydrophobicity increases with incubation time. Furthermore, the nanotube-positive cells were segregated via extensive vortexing at 200 rpm for 15 min. The comparative analysis of planktonic cells and surface-attached nanotube-positive cells indicated a 50% increase in cell hydrophobicity in nanotube-positive cells. Moreover, no significant difference was noticed in small colony variants of *P. aeruginosa* isolates before and after nanotube formation (Figure 10).

## 4. Discussion

In the natural environment, a group of cells may adopt a biofilm lifestyle and adhere to a solid surface [28]. Such a population then invites other cells in its niche to assemble in biofilm consortia and make a strong aggregate [22]. These biofilm consortiums protect indwellers from antibacterial agents and other invaders [27]. Survival in biofilm communities is difficult due to inappropriate food supply, congestion and suffocation [29]. To cope with these challenges, most of the indwellers change their living style, slow down the metabolic machinery, and hold on to multiplication and reproduction [27,29]. Thus, these indwellers can survive and overcome these harsh environmental conditions and maintain persistence for a longer time [29,30]. Moreover, the cells with reduced metabolism are highly adhesive and are generally considered small colony variants [7,31,32].

In the present study, an intriguing property of these colony variants is noticed. These cells colonized the lower base of the consortia and connected via nanotubes. The presence of nanotubes was noticed a few years ago while working on *P. aeruginosa* [22]. Initially, we were unable to properly notice this character and considered it a normal feature of the biofilm population [22]. However, attention was focused as this phenomenon appeared frequently, especially in cells grown under compressed form. A slide was placed accidentally over a firmly adhered biofilm of *P. aeruginosa* on a glass slide. SEM analysis of this particular glass slide displayed a network of cells connected through nanotubes. This phenomenon was confirmed and reproduced by the application of weight (around 5–15 g) on biofilm populations. Interestingly, after the application of ~15.45 g of weight within 6 h, every cell formed nanotubes and connected to neighbouring cells [33]. Have also reported similar kind of findings in a recent report. This finding proposes that the formation of nanotubes is a rapid response of cells under the stress of pressure, and strengthens the notion established by [34] who concluded that nanotubes could tremendously increase the surface area of a cell within minutes and their motion can be traced to a timescale of milliseconds. In the present research, it is also noticed that the application of weight results in the deformation of cell shape. The nanotube-producing cells showed a brick-like appearance. This might be due to the stretching of cell walls to produce nanotubes. This brick shape may also be helpful in the maintenance of internal components of the cell by providing extra space. In all conditions, this cell shape modification seems to be a permanent property of nanotube-producing cells of *P. aeruginosa*. The normal phenotypes of *P. aeruginosa* have not shown this type of shape modification, further validating this belief.

Moreover, cells unable to produce nanotubes were wiped out and the surface was completely occupied by only nanotube producers, indicating a potential selective advantage of these cells over nanotube non-producers. After the removal of weight and restoration of nutrients, these phenotypes reverted to the normal planktonic lifestyle. It indicates that the nanotubes are not merely the phenomenon of dying cells; rather it is a strategy of cells to connect and support each other to allay environmental pressure. It was also supported by the findings of [35], who suggested that nanotubes may be used for both cooperation and competition between cells [35].

The small colony variants (SCVs) offered greater resistance in adherent biofilms to environmental stress as compared to the planktonic ones. It was also endorsed by other studies that SCVs of *P. aeruginosa* are more elastic and adhesive compared to their parent strains and have a major role in the persistence of biofilms [11,36]. However, [11] showed that as biofilms are compressed, cells become stiffer, however, they have targeted general biofilms and our findings are specific to small colony variants of *P. aeruginosa.* Biofilm is a heterogeneous consortium, and it has been reported [37] that some parts of biofilms are elastic solids, some parts are viscoelastic liquids, and the base is stiffer than the upper parts. In this study, it was observed that the SCVs of *P. aeruginosa* may show enhanced elasticity to bear the applied pressure. This increased elasticity may facilitate the development of nanotubes to communicate and share resources among the survivor population. In this context, we agree with the author of a previous study [33], who is of the view that ageing and increased elasticity create weak spots in the cell wall, which may serve as channels through which nanotubes are extruded to relieve intracellular pressure [33].

The other possibility is that the weight pressure may increase the porosity of the cell wall, which may serve as channels through which nanotubes can be extruded. The cells that are incapable of producing nanotubesare unable to sustain the weight pressure and lysed due to loss of cytoplasmic contents. This was further elucidated by the findings of previous studies [38]. Nanotubes maintain intracellular physiological conditions across bacteria attached to a solid surface by sharing resources and information. Additionally, these phenotypes are more hydrophobic with nanotubes, which provide them with an additional factor to sustain for a long time in the adhesive stage. Another characteristic feature was the site of nanotube formation. The majority of the nanotubes of *P. aeruginosa* were positioned at the cell poles. One study [39] reported that poles are locations for the assembly of surface organelles of rod-shaped bacteria to control directional motility and surface adhesion. They are used for cell-to-cell interaction, surface adhesion and sharing of cytoplasmic contents [38]. The phenomenon of developing nanotubes was observed at the stationary phase in cells directly attached to a solid surface [40]. It has been reported that under nutritional depletion bacteria interact through nanotubes for the exchange of nutrients. This statement is also in concordance with our findings. The subject isolates were found to produce nanotubes to overcome stress by connecting and sharing resources in an organized system. The mid of the cell connections were also noticed in cases where one cell was connected to more than one neighbour. In this situation it is hypothesized that the preferred positions of nanotubes were polar in cells and due to a lack of attachment sites, they may have made contact at mid of the cell. Additionally, previous studies reported inter- and intra-species nanotubes [33,38] However, the present studies showed only intra-species connections through the nanotube. Three different isolates i.e., *P. aeruginosa*, *E. coli* and *S. aureus* were used to find inter-species nanotubes. All of these isolates showed intra-species nanotubes, but no inter-species connection was found. Furthermore, these are obligate connectivity tools because no free or without intra-species connection nanotubes were found at any stage. These characters distinguish nanotubes from other bacterial appendages. In this regard, our study is in agreement with previous studies [41].

The *mreB* and *oprF* expression indicate the cells are active and regulate proper cell growth [42]. It has been suggested that *MreB* is highly conserved among rod-shaped bacteria and essential for growth under normal growth conditions. *MreB* directs the localization of cell wall synthesis and loss of *MreB* results in round cells and death. Similarly, *OprF* is involved in several crucial functions, including cell structure, outer membrane permeability, environmental sensing, and virulence [43]. Furthermore, *OprF* is orthologous to OmpA of Enterobacteriaceae. *OprF* anchors the outer membrane to the peptidoglycan layer. It is one of the determinant factors of virulence and is involved in host–pathogen interaction. The expression of *OprF* is controlled by the same regulatory factors involved in the homeostasis of the cell envelope and the regulation of biofilm phenotypes. One study [44] reported that a reduction in *OprF* expression coincided with increased biofilm, elevated EPS and cellular aggregation. This expression variation of these two genes strongly suggested that most of the cells in sonicated suspensions of adherent biofilms were members of biofilm consortia and not planktonic ones. As explained previously [44,45], the regulation of *OprF and mreB* leads to increased formation of biofilm and pel-exopolysaccharide. The *mreB* is fundamentally involved in the regulation of cellular growth, morphogenesis, and cell division. Biofilms developed under weight pressure slides consisted of live cells that rejuvenated to growth upon sub-culturing on microbial media. This finding provides a strong rationale for the belief that the formation of nanotubes under the stress of pressure is not merely the expressive trait of dying cells as suggested earlier; rather it may represent another mechanism of the resistance of bacteria to environmental stress [33]. It further supported the hypothesis that nanotubes are not associated with cell death, but a strategy to survive. The application of Triton X-100 resulted in the complete removal of nanotubes and disruption of biofilm consortia. Although the cells were squeezed and died, the basic structure was intact in a rod shape. It suggests that Triton X-100 is disrupting bacterial membranes and other lipid-made structures without damaging cell wall integrity. It has also been reported [46] that Triton X-100 was effective for the removal of the cytoplasmic membrane without damaging the normal cell wall morphology. Triton X-100 is a lipid solvent. A recent report [47] mentioned that bacterial nanotubes are made of lipids and may have an important role in bacterial communication. In biofilm consortia, these tubes have a critical role and are also an easy target due to their composition and their solubility in lipid solvents.

## Figures and Tables

**Figure 1 cells-11-03374-f001:**
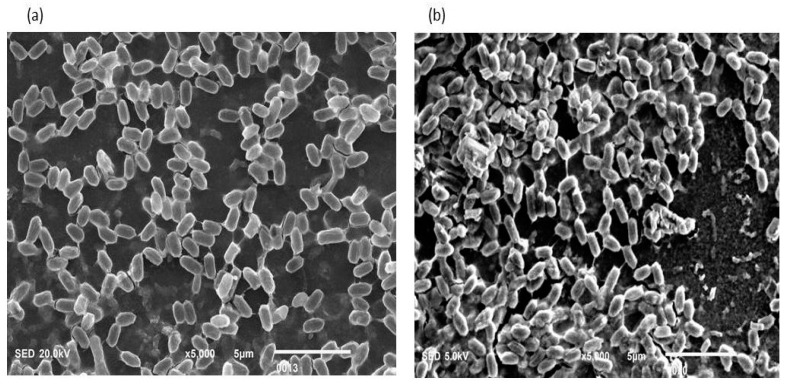
(**a**,**b**): Biofilm-positive cells occupying the lower base of biofilm consortia and adhered to the glass surface. The weight was not applied to these cells. Therefore, the cells maintained a typical *P. aeruginosa* shape (Rod shape). The majority of cells are scattered and connected side by side. However, a small population is also connected through nanotubes.

**Figure 2 cells-11-03374-f002:**
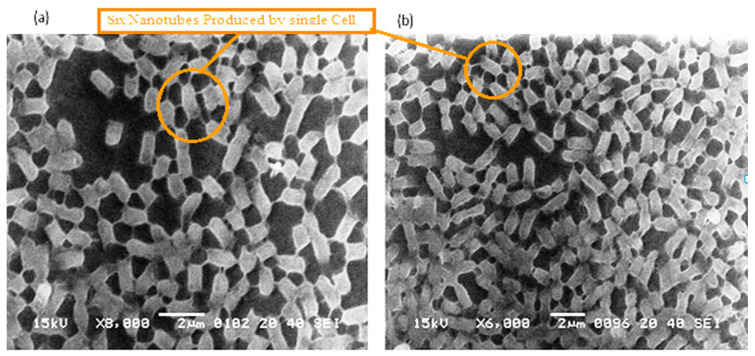
(**a**,**b**): These cells also occupied the lower base of biofilm consortia and adhered to the glass surface. This population was grown in between two slides under a weight of (15.45 g). Due to the stress of weight, cells were unable to maintain the typical *P. aeruginosa* shape (rod shape). However, these cells developed an interconnected network of nanotubes and covered the whole surface in a mesh-like structure.

**Figure 3 cells-11-03374-f003:**
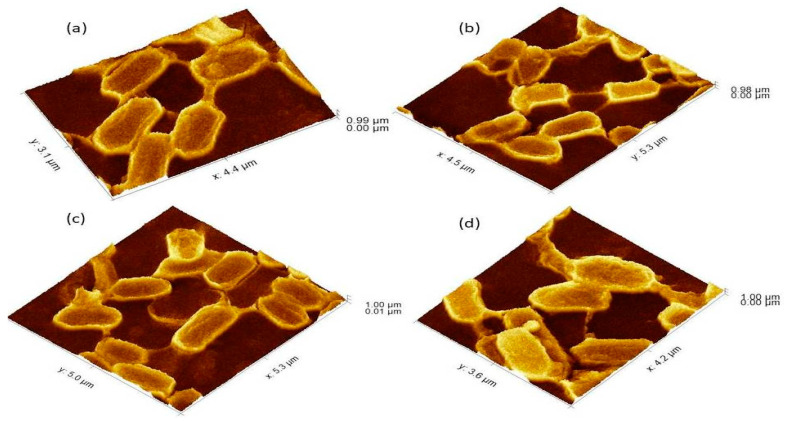
The 3Drepresentation of biofilm-positive cells occupying the lower base of biofilm consortia and adhered to the glass surface. The weight was not applied to these cells, therefore, the cells maintained typical *P. aeruginosa* morphology i.e. rod shape. (**a**): Cells with normal *P. aeruginosa* morphology. (**b**): In this 3D image, the formation of nanotubes can be visualized connecting surface-attached cells of adherent biofilms. (**c**): cells connected Side by side as well as through nanotube. (**d**): Extracellular matrix material attached to the cell surface.

**Figure 4 cells-11-03374-f004:**
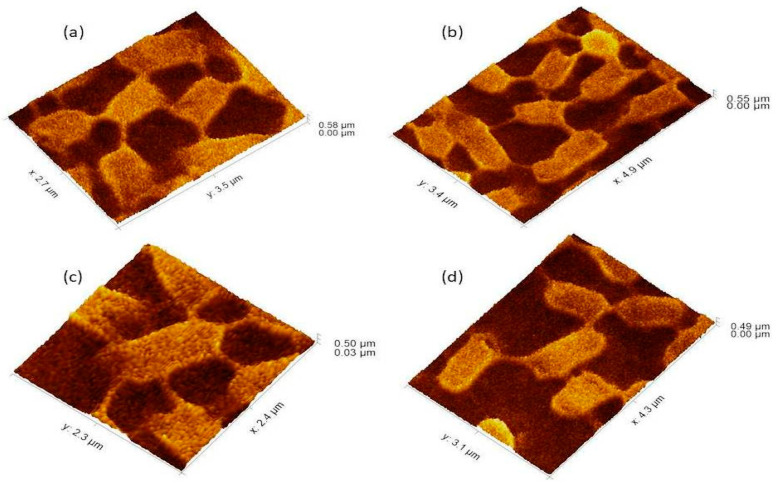
The 3Drepresentation highlighted the cells that occupied the lower base of biofilm consortia and adhered to the glass surface. This population was grown between two slides under a weight of (15.45 g). Due to the weight pressure, cells were unable to maintain the typical *P. aeruginosa* cylindrical rod shape. (**a**): Cells with a compressed shape (Brick-like appearance) due to weight pressure. (**b**): Heterogeneous (large and small rod) cells attached to the glass surface and connected through a nanotube. (**c**): A single cell connected to multiple (6) neighbours through a nanotube. (**d**): Cells connected to one or two neighbours through a nanotube.

**Figure 5 cells-11-03374-f005:**
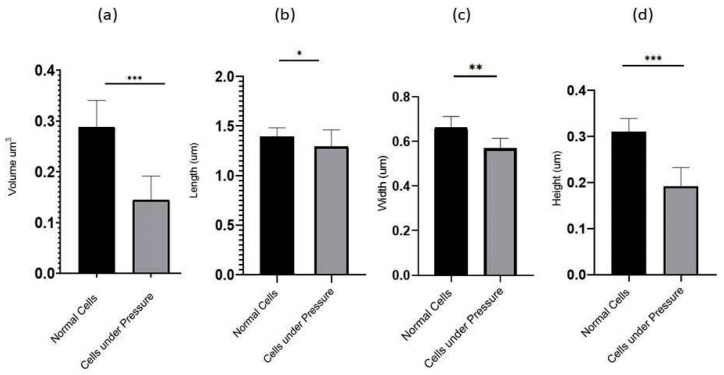
The graph depicted the compression of Volume (**a**) and physical dimensions: length (**b**), Width (**c**), and Height (**d**) between 35 randomly selected cells under the stress of weight (15.45 g) and normal cells. The comparative analysis indicated that the cells under weight were reduced in volume. Conversely, cells occupying the lower base of biofilm consortia without weight pressure (normal cells) maintained the normal physiology of *P. aeruginosa*. All parameters were measured from SEM and AFM images using Gwyddion 2.60. All values are statistically significant, *, **, *** represent *p* values <0.002, *p* < 0.001 and *p* < 0.0001, respectively.

**Figure 6 cells-11-03374-f006:**
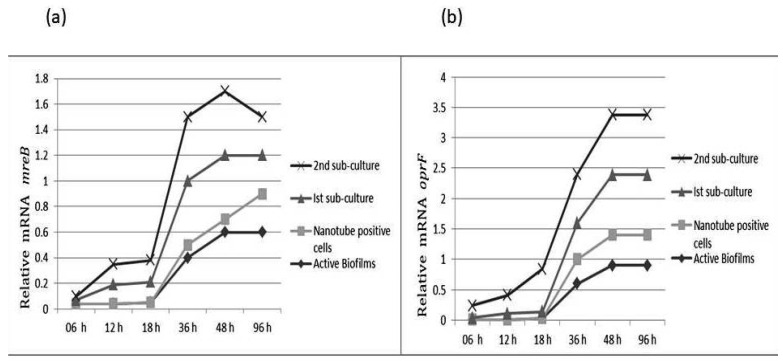
Relative gene expression of *mreB* (**a**) and *oprF* (**b**) of subject isolates of *P. aeruginosa* at different growth phases and at different time intervals.

**Figure 7 cells-11-03374-f007:**
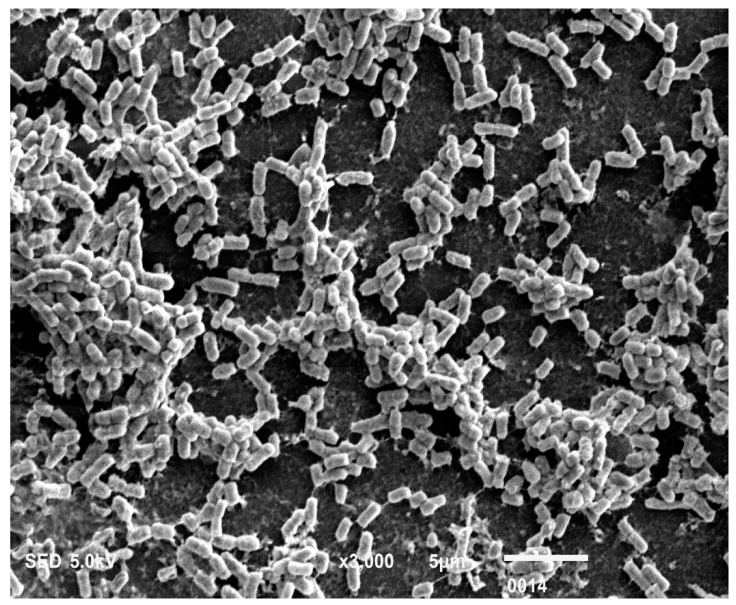
Free-floating multicellular aggregates or clumps of cells collected from broth tube. The clumps or multicellular aggregates formed in broth, especially at the air-water interface after 48 h of incubation. These floating multicellular aggregates do not adhere to any surface and the majority of the cells are devoid of nanotube structure.

**Figure 8 cells-11-03374-f008:**
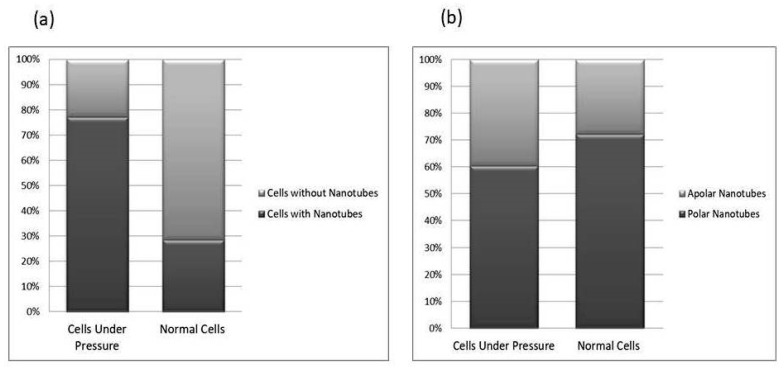
This comparative study of nanotube possession in cells. Image (**a**) reflects the total number of nanotubes in cells grown in between two slides under a weight (15.45 g) and cells from the lower base of biofilm consortia without weight pressure (normal cells). The cells under-weight pressure possesses more nanotubes as compared to cells without pressure. The (**b**) side of the figure depicted the position of nanotubes on cells. The majority of cells in both conditions possess polar nanotubes. Calculations were made on Gwyddion 2.60 using SEM and AFM data of more than 350 cells in total; all experiments were performed in triplicate.

**Figure 9 cells-11-03374-f009:**
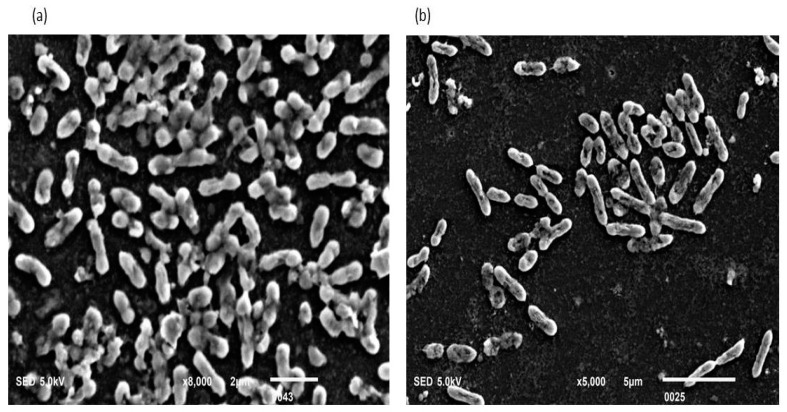
(**a**,**b**) The biofilm consortia were exposed to Triton X100 for 30 s. The picture depicted that the consortium was dispersed in 30 s, and the cells were killed and squeezed. The extracellular matrix material was wiped and nanotubes disappeared from the cell surface. The results indicated that Triton X100 was equally effective against biofilm consortia developed in between two slides under-weight pressure and against normal biofilms.

**Figure 10 cells-11-03374-f010:**
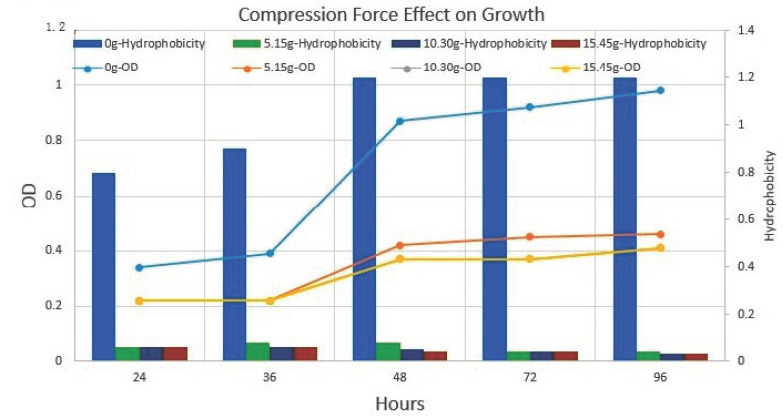
Compression force (weight) effect on growth rate and cell surface hydrophobicity. The bar (secondary *Y*-axis) in the graph is representing cell surface hydrophobicity. The line (Primary *Y*-axis) in the graph is representing cell growth at different weight pressures and without weight. The results indicate the application of force resulted in a slowdown of growth rate and hydrophobicity. Nanotube in mix culture of *P. aeruginosa*, *E**. coli* and *S. aureus*.

**Table 1 cells-11-03374-t001:** Physical Measurements (Total volume including cell length, width and height) of 35 randomly selected cells from a biofilm population of 350 cells. These cells were attached to the glass surface at the lower base of the biofilm consortia. All estimates were analyzed using Gwyddion 2.60.

Normal Cells and Nanotube Measurements										
Normal Cells Measurements							Nanotubes Measurements			
Length (um)	Width (um)	Height (um)	Volume (um^3^)	Nanotubes	Polar	A-Polar	Length (nm)	Width (nm)	Height (nm)	Volume (um^3^)
1.53	0.66	0.32	0.323	1	1	0	545	163	162	0.0143
1.41	0.68	0.29	0.278	0	0	0				
1.37	0.71	0.33	0.320	2	1	1	* 704	*143	* 118	0.0118
1.41	0.64	0.29	0.261	0	0	0				
1.47	0.61	0.31	0.277	0	0	0				
1.44	0.59	0.28	0.237	2	1	1	* 714	* 117	* 169	0.0141
1.58	0.71	0.34	0.381	0	0	0				
1.49	0.59	0.30	0.263	0	0	0				
1.47	0.58	0.31	0.264	0	0	0				
1.37	0.72	0.35	0.345	0	0	0				
1.36	0.64	0.34	0.295	0	0	0				
1.33	0.67	0.32	0.285	0	0	0				
1.57	0.74	0.38	0.441	4	3	1	* 946	* 155	* 206	0.0302
1.58	0.62	0.33	0.323	0	0	0				
1.34	0.72	0.31	0.299	0	0	0				
1.30	0.71	0.33	0.304	1	1	0	442	98	222	0.0096
1.34	0.67	0.29	0.260	0	0	0				
1.36	0.69	0.29	0.272	1	1	0	336	231	276	0.0214
1.33	0.67	0.25	0.222	0	0	0				
1.26	0.64	0.27	0.217	0	0	0				
1.35	0.64	0.29	0.250	0	0	0				
1.33	0.68	0.31	0.280	3	2	1	* 454	* 94	* 168	0.0071
1.25	0.66	0.29	0.239	0	0	0				
1.42	0.76	0.34	0.366	0	0	0				
1.22	0.61	0.28	0.208	0	0	0				
1.37	0.72	0.33	0.325	0	0	0				
1.35	0.74	0.33	0.329	0	0	0				
1.37	0.62	0.26	0.220	0	0	0				
1.39	0.64	0.32	0.284	2	1	1	* 592	* 121	* 314	0.0224
1.33	0.61	0.31	0.251	1	1	0	756	183	258	0.0356
1.39	0.66	0.28	0.256	0	0	0				
1.41	0.73	0.37	0.380	0	0	0				
1.36	0.61	0.31	0.257	0	0	0				
1.38	0.61	0.29	0.244	0	0	0				
1.52	0.63	0.32	0.306	1	1	0	488	147	128	0.0091
* 1.392	* 0.662	* 0.310	* 0.288	** 18	72.2%	27.8%	* 597.7	* 145.2	* 202.1	* 0.005

* Represent mean, ** represent total count.

**Table 2 cells-11-03374-t002:** Physical Measurements (Total volume including cell length, width and height) of 35 randomly selected cells from a biofilm population of 350 cells. These cells were recovered from biofilm consortia developed in between two slides under a weight of (15.45 g). All estimates were analyzed using Gwyddion 2.60.

Cells under Pressure and Nanotube Measurements										
Cells under Pressure Measurements							Nanotubes Measurements			
Length (um)	Width (um)	Height (um)	Volume (um^3^)	Nanotube/s	Polar	A-Polar	Length (nm)	Width (nm)	Height (nm)	Volume (um^3^)
1.16	0.54	0.14	0.087	4	3	1	* 458	* 144	* 106	0.006
1.35	0.52	0.19	0.133	0	0	0				
1.12	0.52	0.13	0.075	2	1	1	* 488	* 152	* 60	0.004
1.54	0.63	0.21	0.203	1	1	0	431	198	85	0.007
1.61	0.63	0.19	0.192	4	2	2	* 411	* 144	* 116	0.006
1.17	0.62	0.19	0.137	1	1	0	254	162	97	0.003
1.51	0.61	0.22	0.202	4	3	1	* 429	* 187	* 86	0.006
1.24	0.59	0.19	0.139	4	1	3	* 376	* 106	* 100	0.003
1.33	0.61	0.17	0.137	4	3	1	* 433	* 147	* 60	0.003
1.17	0.53	0.16	0.099	0	0	0				
1.26	0.59	0.15	0.111	4	2	2	* 185	* 147	* 128	0.003
1.24	0.61	0.19	0.143	3	1	2	* 588	* 142	* 98	0.008
1.36	0.62	0.17	0.143	1	0	1	765	164	136	0.017
1.14	0.61	0.17	0.118	0	0	0				
1.01	0.61	0.16	0.098	2	1	1	* 478	* 147	* 94	0.006
1.17	0.53	0.12	0.074	4	2	2	* 473	* 114	* 82	0.004
1.23	0.63	0.24	0.185	4	2	2	* 485	* 123	* 64	0.003
1.18	0.57	0.19	0.127	3	1	2	* 491	* 133	* 88	0.005
1.32	0.55	0.18	0.130	1	1	0	522	139	74	0.005
1.42	0.55	0.17	0.132	1	1	0	261	145	107	0.004
1.45	0.57	0.22	0.181	0	0	0				
1.56	0.63	0.27	0.265	4	2	2	* 421	* 122	* 94	0.004
1.15	0.51	0.14	0.082	0	0	0				
0.97	0.54	0.22	0.115	1	1	0	379	119	102	0.004
1.37	0.61	0.25	0.208	4	3	1	* 428	* 115	* 93	0.004
1.34	0.57	0.19	0.145	1	1	0	234	126	114	0.003
1.05	0.51	0.25	0.133	3	3	0	* 451	* 132	* 79	0.004
1.54	0.55	0.26	0.220	0	0	0				
1.36	0.55	0.26	0.194	0	0	0				
1.41	0.62	0.25	0.218	2	1	1	* 383	* 104	* 84	0.003
1.04	0.54	0.15	0.084	4	2	2	* 394	* 127	* 99	0.004
1.33	0.5	0.17	0.113	3	2	1	* 526	* 128	* 81	0.005
1.18	0.58	0.17	0.116	3	2	1	* 417	* 135	* 96	0.005
1.36	0.51	0.22	0.152	0	0	0				
1.58	0.51	0.18	0.145	4	3	1	* 438	* 128	* 116	0.006
* 1.292	* 0.570	* 0.192	* 0.144	** 76	60.52%	39.4%	* 429.5	* 138.1	* 94.03	* 0.004

* Represent mean, ** represent total count.

## Data Availability

Not applicable.

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
