# Peer review of "Nanotubes Formation in P. aeruginosa"

_cells, 2022, doi:10.3390/cells11213374_

Round 1

Reviewer 1 Report

The manuscript “Nanotubes formation in P. aeruginosa” by Ahmed et al, show the study of nanotube formation in P. aeruginosa grown under different conditions (temp, pH and applied weight).

Major comments:

- the presentation of the data is very disorganized and not well described along the manuscript

- Figure legends are very short and not well presented, information is missing.

- High magnification SEM images would be ideal to appreciate some of the nanotubes described along the manuscript.

- Authors should include image and e-beam conditions utilized to acquired EM data.

- Authors need to justify the use of different landing energies utilized in the various SEM images presented in the manuscript.

- Authors should provide a better description of how biofilms were dehydrated for SEM data acquisition and analysis since the selected protocol could affect the final outcome of their quantifications.

- It is not clear what the authors mean by “brick-like appearance” (Fig 2). If the cellular shape is affected by the grow conditions is this only a feature related to cellular stress? A better description and discussion are needed to support their findings.

- Did the authors collected tilted images to obtain the cellular volume data? How was the volume of cells quantify without any z information?

- AFM data is not well described in the manuscript, how is this data complementing the results obtained by SEM?

- The section “Nanotube in Mix Culture of P. aeruginosa, E. coli and S. aureus” is described as “Data Not Published”. Since many other researchers in the field have shown the presence and formation of intra-species nanotubes for some of these species the authors should discuss the lack of these particular nanotubes in their experimental design.

Minor comments:

- page 5 line 197 replace “week” for “weak”

- Figure 3 is located at the end of the manuscript for some reason and this needs to be corrected. The data provided with Triton X100 is somehow irrelevant since this strong detergent is expected to affect biofilm formation as well as bacterial survival.

- Figure 11 is included in the text prior to Figure 10.

- Figure 10. OD (bacterial growth) should be in log scale.

Author Response

I am pleased and obliged to submit the revised draft of the manuscript entitled “Nanotubes formation in P. aeruginosa. I am deeply thankful to you and the esteemed reviewers for their insightful feedback. We have put our best efforts to entertain all the suggestions. Changes made have been highlighted in the revised manuscript.

Reviewer 2 Report

This manuscript by Ahmed et al presents a study that involved the development of nanotubes in P.aeruginosa. They reported that the nanotubes were observed at the stationary phase of biofilm indwellers and were more prominent after applying weight to consortia. Overall, the current data to support their findings is acceptable whereas there are still some issues that need to be addressed.

1. The Introduction should provide the background and state the objectives of the work, avoiding a detailed literature survey. I cannot understand the significance of this work from the present introduction. In my view, this section should be intensively polished.

2. Formation of Biofilm: “washed with Phosphate buffer saline (PBS) (pH=7.0)” pH = 7.0? Why do you use PBS with 7.0, but not 7.4? “Subsequently, 100μL of the above cell suspension was inoculated in each flask.” What is the density of P. aeruinosa in the cell suspension?

3. “It was observed that a single cell may produce as much as six nanotubes and could connect simultaneously to six neighbors in different directions (Figure 6c).” It seems that Figure 6c can’t support this claim. More representative images should be provided to verify this point.

4. “Similarly, colonies picked from the agar surface after 120 h of incubation didn’t show noticeable nanotube formation.” The corresponding results should be provided. And please clarify why a long time incubation didn’t show noticeable nanotubes formation?

5. Figure 4: The authors asserted that these cells were devoid of nanotube structure. Nevertheless, it looks some visible nanotube structures can be found in this image.

6. Figure 7b: In my experience, I don’t think there is a statistically significant between cells under pressure and cell at normal atm in length. It is weird that the authors get the P value < 0.002. I recommend rechecking the original data and the statistical results.

7. Effect of temperature change on nanotube formation: I can’t find any data in this section. At least some relevant data or images should be provided to support the claims here.

Author Response

(The authors gave the same response as above.)

Round 2

Reviewer 1 Report

- AFM description is still missing from the Materials and Methods section.  This section has not been updated in the revised manuscript as requested by this reviewer. 

- Authors missed to include imaging acquisition parameters and e-beam conditions utilized to acquired SEM data. There is no justification for the big differences in the landing energies utilized in the various SEM images presented along the manuscript. Was this needed due to the nature of the specimens? Is this somehow relevant to their findings?

Author Response

Respected Reviewer,                                              Dated: 5th Oct, 2022

Cells

I am pleased and obliged to submit the (2nd Round) revised draft of the manuscript entitled “Nanotubes formation in P. aeruginosaaccepted for publication in Cells. I am deeply thankful to you for your insightful feedback. We have put our best efforts into entertaining all the suggestions. Changes made have been highlighted in the revised manuscript.

  • Scanning Electron Microscopy (SEM) facility provided by Dow University of Health Sciences Dental College Karachi, Pakistan.
  • Expertise is provided by the lab Engineer and we follow his instruction for imaging. We are sorry for the late response because Scanning Electron Microscopy is not available in our lab.
  • English grammar and minor technical errors have also been improved in a (2nd Round) revised draft of the manuscript.
  • If you still feel any confusion then you can feel free to contact us, we will try again our best to reply to you. We are facing flood issues in our country that’s why taking time as well.

Reviewer 2 Report

I have no more comments.

Author Response

Dear Reviewer,                                                            Dated: 5th Oct, 2022

Cells

I am pleased and obliged to submit the (2nd Round) revised draft of the manuscript entitled “Nanotubes formation in P. aeruginosa” accepted for publication in Cells. I am deeply thankful to you for your insightful feedback. We have put our best efforts into entertaining all the suggestions. Changes made have been highlighted in the revised manuscript.

  • Scanning Electron Microscopy (SEM) facility provided by Dow University of Health Sciences Dental College Karachi, Pakistan.
  • Expertise is provided by the lab Engineer and we follow his instruction for imaging. We are sorry for the late response because Scanning Electron Microscopy is not available in our lab.
  • English grammar and minor technical errors have also been improved in a (2nd Round) revised draft of the manuscript.
  • If you still feel any confusion then you can feel free to contact us, we will try again our best to reply to you. We are facing flood issues in our country that’s why taking time as well.